# An Image is Worth $K$ Slots: Data-efficient Scaling of Self-supervised Visual Pre-training

## Abstract

Scaling up data and computing has become the norm for pre-training powerful visual encoders. Current algorithms, when scaled up, often require training on large-scale datasets that are unlikely to be object-centric. However, these algorithms were typically developed and validated on the object-centric ImageNet. This discrepancy may suggest sub-optimal scalability and underutilized data potential. Non-object-centric (NOC) data, with its multiple objects and complex layouts, tends to be more information-dense. To better leverage this underlying structure, we introduce a semantic bottleneck to MIM, which reduces the number of prototypes to encourage the emergence of objectness at patch-level token representation. Further, cross-view consistency regularization is applied to encourage multiview invariance. Together, this induces semantic object discovery and allows instance discrimination to be applied between object-level features (slots). Our experiments encompass pre-training on object-centric, scene-centric, web-crawled, and ego-centric data. Across all settings, our approach learns transferrable representations and achieves significant improvements over prior work in image recognition, scene understanding, and robot learning evaluations. When scaled up with million-scale datasets, our method also demonstrates superior data efficiency and scalability. We will make our code and model artifacts publicly available.

## 1 Introduction

Self-supervised representation learning from visual data has seen significant progress, evolving from contrastive learning (Chen et al., 2021; Caron et al., 2021) to masked image modeling (MIM) (Bao et al., 2022; He et al., 2022; Xie et al., 2022; Wei et al., 2022) and hybrid methods (Zhou et al., 2022; Oquab et al., 2024), which have benefited numerous downstream tasks. A key advantage of self-supervised learning is its ability to learn representations from unlabeled data, eliminating the need for human annotations and making it easier to scale up training datasets. Despite this advantage in utilizing diverse types of data, most research has focused on object-centric datasets like ImageNet for model development, leaving large volumes of non-object-centric (NOC) data, such as Open Images (Kuznetsova et al., 2020), SA-1B (Kirillov et al., 2023), LAION (Schuhmann et al., 2022), and Ego4D (Grauman et al., 2022), underutilized. However, many primary application domains of self-supervised learning – such as object detection, image segmentation, and robotics – often require handling NOC data.

This motivates us to explore the potential of NOC data for self-supervised learning, which is rich in information, offers new opportunities for data scaling, and could bridge the data-domain gap between self-supervised learning and real-world applications. While some research has investigated scene-centric data for self-supervised dense representation learning by developing pixel-level (Xie et al., 2021; Wang et al., 2021; Zhou et al., 2022) and object-level (Hénaff et al., 2021; 2022; Wen et al., 2022) contrastive learning objectives using learned or handcrafted objectness, these studies have primarily relied on ResNet-based backbones. As a result, it remains unclear how well these methods translate to modern architectures like vision transformers (see Tab. 1). Although one might argue that the state-of-the-art self-supervised model, DINOv2 (Oquab et al., 2024), already utilizes NOC data with a vision transformer backbone, its success heavily depends on data curation techniques that leverage the object-centric ImageNet dataset to select neighboring data, keeping its data distribution closely tied to object-centric approaches. Our preliminary experiments also suggest unsatisfactory results of DINOv2 using the same NOC data setup (Figs. 3 and 5).

To this end, we begin by conducting a comprehensive evaluation of existing self-supervised learning approaches on four datasets: object-centric (Deng et al., 2009) and non-object-centric (Lin et al., 2014; Changpinyo et al., 2021; Grauman et al., 2022). While the performance of current methods on non-object-centric data is suboptimal from a representation learning perspective, our study reveals that several insights from object-centric learning remain applicable to NOC data. Specifically, cross-view learning (Tian et al., 2020b) encourages semantic learning by enforcing feature invariance to data augmentations, while MIM, suitable for pre-training transformer-based architectures, is particularly effective at capturing fine-grained, low-level representations (see Fig. 1 for visualizations).

Motivated by these insights, we propose SlotMIM, a method that repurposes and integrates masked image modeling (MIM) and contrastive learning for effective representation learning from NOC datasets. The core idea of SlotMIM is to group patch-level image tokens into object-level feature abstractions, referred to as "slots", thereby decomposing NOC data into object-centric slots so that object-centric techniques can be effectively applied. To make patch-level tokens more semantically aware for subsequent grouping, we enhance MIM with cross-view consistency regularization. Additionally, we introduce a semantic bottleneck, which reduces the number of prototypes to encourage the emergence of semantic and objectness at patch-level token representations (see Fig. 1). Building on these semantically enriched patch tokens, we apply attentive pooling over the learned patch-level features, using prototypes to initialize object representations, thereby grouping patches into object-level slots and decomposing an image into object representations. Contrastive learning (Chen et al., 2021) is further applied to these slots to promote the discriminativeness of the learned representations. Together, these designs enable us to perform effective representation learning from NOC data.

In our experiments, we pretrain on a diverse range of datasets, including object-centric, scene-centric, web-crawled, and ego-centric data. We evaluate the pre-trained models on various tasks such as ImageNet linear probing and fine-tuning, ADE20K semantic segmentation, COCO detection/instance segmentation, and visuomotor control for robotics. Across all these evaluations, our method consistently outperforms existing approaches by a significant margin, showing 1) Data efficiency: It maximizes the utility of available data, reducing the dependency on continually scaling up data collection; 2) Domain adaptability: SlotMIM shows superior adaptability to datasets that are closer to the downstream application domains and potentially richer in information; and 3) Scalability: The method not only performs well at smaller scales but also scales efficiently with increasing data size.

In summary, our contributions can be summarized as follows:

- We conducted a comprehensive revisiting study across three non-object-centric datasets. Our findings reveal that non-object-centric (NOC) data is rich in information with vast potential, yet it remains underutilized in current approaches (Secs. 3.2 and 3.3).

- We formalize representation learning from NOC data into two key sub-tasks: decomposition and object-centric representation learning. By repurposing established techniques to target these specific sub-tasks, we developed a unified approach that effectively works for both non-object-centric and object-centric data, offering a robust solution that bridges the two domains (Fig. 3).

- Our method maximizes data utilization, achieving both data efficiency and excellent scalability. This contributes to the field of pre-training by exploring a new avenue for scaling up models using NOC data. At the same time, our approach delivers pre-trained models that are better suited for downstream tasks, including robotics, providing more relevant and effective solutions for real-world applications (Secs. 3.4 and 3.5).

## 2 METHOD

### 2.1 PRELIMINARIES

**Deep clustering as self-distillation.** DINO (Caron et al., 2021) is a discriminative self-supervised learning approach that learns a set of $C$ prototypes online that clusters image embeddings. Let $\boldsymbol{x} \in \mathbb{R}^{H \times W \times 3}$ be an input image, and $f_\theta, f_\xi : \mathbb{R}^{H \times W \times 3} \to \mathbb{R}^d$ be student and teacher encoder networks parameterized by $\theta$ and $\xi$ respectively. Let $\boldsymbol{z}_\theta = f_\theta(\boldsymbol{x})$ and $\boldsymbol{z}_\xi = f_\xi(\boldsymbol{x})$ denote the embeddings produced by the student and teacher networks (we omit the projector for simplicity). The cluster assignments are computed as $p_\theta(\boldsymbol{x}) = \mathrm{softmax}(\boldsymbol{z}_\theta \cdot \mathcal{C}/\tau)$, where $\mathcal{C} = \{\boldsymbol{c}_c\}_{c=1}^C$ are the

prototypes, and $\tau$ is a temperature parameter. Then the loss is computed as cross-entropy between predictions of student model and teacher model: $\mathcal{L}_{\text{DINO}}(\boldsymbol{v}^1, \boldsymbol{v}^2) = -\sum_{c=1}^C q_\xi(\boldsymbol{v}^2)_c \log p_\theta(\boldsymbol{v}^1)_c$, where $\boldsymbol{v}^1$ and $\boldsymbol{v}^2$ are two augmented views of the same image. Since it resembles knowledge distillation with soft labels produced by the model itself, it is also dubbed as self-distillation.

**DINO on image patches with MIM.** iBOT (Zhou et al., 2022) extends the DINO objective from global image embeddings to local image patches with masked image modeling (MIM). Let $\mathcal{M} \in \{0, 1\}^N$ be a binary mask indicating which patches are masked, where $N$ is the total number of patches. The masked input $\tilde{\boldsymbol{v}}$ is defined as $\tilde{\boldsymbol{v}}_i = \boldsymbol{m}$ if $\mathcal{M}_i = 1$, and $\tilde{\boldsymbol{v}}_i = \boldsymbol{v}_i$ otherwise, where $\boldsymbol{m}$ is a mask token. The iBOT loss predicts the clustering assignments of masked patches given unmasked patches: $\mathcal{L}_{\text{iBOT}}(\boldsymbol{v}) = \sum_{i:\mathcal{M}_i=1} \mathcal{L}_{\text{DINO}}(\tilde{\boldsymbol{v}}_i, \boldsymbol{v}_i)$, where $\tilde{\boldsymbol{v}}_i$ is the masked patch from the student model and $\boldsymbol{v}_i$ is the corresponding unmasked patch from the teacher model.

**Slot attention** (Locatello et al., 2020) is a variant of cross-attention that normalizes attention scores on the query side instead of the key side, introducing competition between queries and encouraging them to focus on different parts of the input. Our approach performs attentive pooling on patch embeddings according to their clustering assignments, sharing high-level intuition with slot attention if viewing the prototypes $\mathcal{C}$ as queries and patch embeddings $\boldsymbol{z}_{\theta,i} = f_\theta(\boldsymbol{x}_i)$ as keys. We thus follow tradition and call the pooled object features slots – prototypes adapted to image patches.

## 2.2 MOTIVATION AND FRAMEWORK

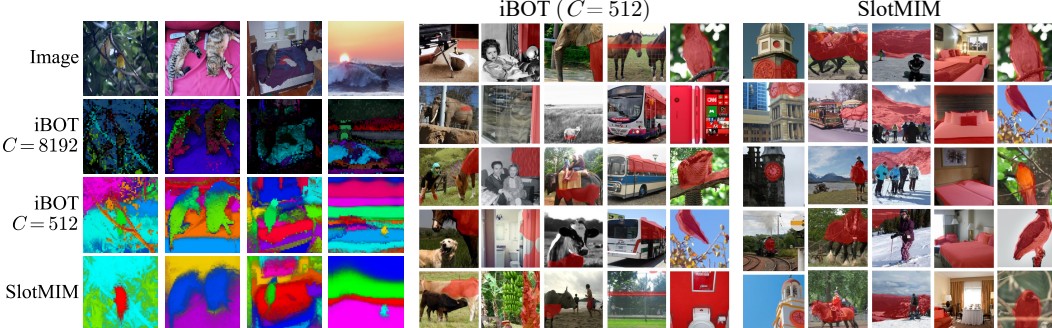

(a) Clustering assignment of patch tokens.     (b) Top-5 segments retrieved by the prototypes (by column).

Figure 1: **Comparison of concepts learned by iBOT and SlotMIM.** iBOT's prototypes can discover fine-grained patterns, and the quality improves if a smaller vocabulary is used (left). But these patterns are bottom-up and lack semantic meaning. In contrast, concepts with same tokens of SlotMIM are semantically coherent and more suitable for building instance discrimination pretext tasks (right).

**High-level intuition.** We decouple self-supervised learning on non-object-centric data into two subtasks: 1) learning to group image patches into objects (or stuff); and 2) learning to discriminate objects as previous works have done on object-centric data. The major challenge here is unsupervised object discovery, which we find could emerge from iBOT with a smaller number of prototypes.

**Representation bottleneck induces objectness from iBOT.** We first investigate the prototypes of iBOT, which is a set of embeddings $\mathcal{C} = \{\boldsymbol{c}_c\}_{c=1}^C$ that clusters image patches into $C$ clusters and assigns each patch token a soft one-hot encoding $p_\theta(\boldsymbol{x}_i)$ identifying its clustering assignment. Conventionally, $C$ is set to be 8192 to capture fine-grained patterns, which is good for learning representations. But in our case, the role of representation learning would be taken by another objective (contrastive learning between slots) and the prototypes are designated to focus on object discovery. We find a much smaller $C$, *e.g.*, 512 for COCO, would suit this goal better because it can build a very compact information bottleneck that forces the model to learn highly compositional concepts – objects. As shown in Fig. 1a, the clusters discovered by iBOT are very fine-grained (2nd row), and objectness emerges if a small vocabulary is used (3rd row). However, these patterns still lack semantic meaning and could split the same object into multiple parts. Also, it remains hard to match discovered objects between views as their semantics vary a lot despite having the same token (Fig. 1b, left). Both issues call for a set of semantic-level prototypes.

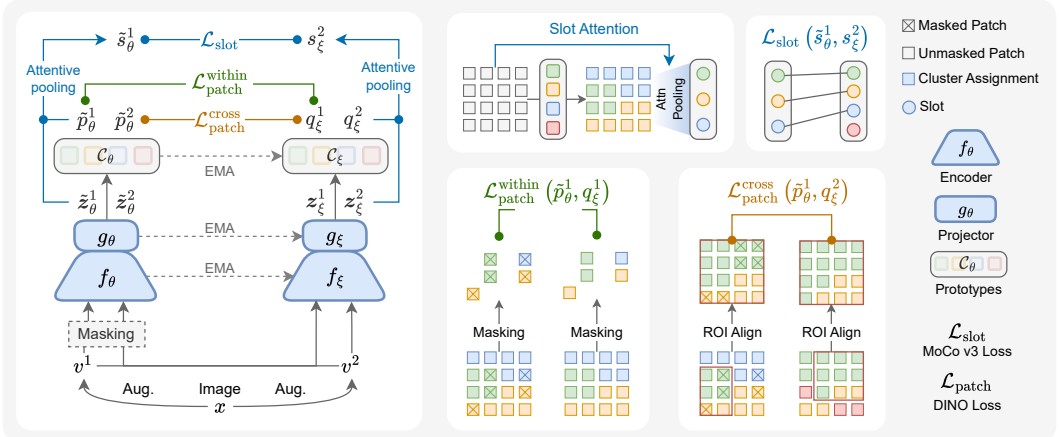

Figure 2: **Overview of SlotMIM.** We repurpose iBOT's within-view patch-level loss for object discovery, add a cross-view objective for semantic guidance, and build object-centric contrastive learning on top of object features (slots) grouped from patches with identical clustering assignments.

**Cross-view consistency lifts object discovery to semantic-level.** A key factor contributing to the lack of semantic meaning is that the iBOT loss $\mathcal{L}_{\text{iBOT}}$ is computed between patches within the same view. Consequently, there is no explicit guidance for learning invariant representations across different views of the same object or scene. We apply a simple yet effective fix: adding a cross-view consistency objective $\mathcal{L}_{\text{patch}}^{\text{cross}}$ that enforces patches undergone different photometric and geometric transformations to have the same token. To match patches between views, we adopt a SlotCon-style mechanism that crops & resizes the overlapping regions of two views (using ROIAlign). Formally, let $\boldsymbol{v}^1$ and $\boldsymbol{v}^2$ be two augmented views of the same image, and $\tilde{\boldsymbol{z}}_{\theta,i}^1 = f_\theta(\tilde{\boldsymbol{v}}_i^1)$ and $\boldsymbol{z}_{\xi,j}^2 = f_\xi(\boldsymbol{v}_j^2)$ be the corresponding patch embeddings. The cross-view consistency loss is defined as:

$$\mathcal{L}_{\text{patch}}^{\text{cross}}(\boldsymbol{v}^1, \boldsymbol{v}^2) = -\frac{1}{|\mathcal{P}|} \sum_{(i,j) \in \mathcal{P}} \sum_{c=1}^{C} \boldsymbol{q}_{\xi,i,c}^2 \log \tilde{\boldsymbol{p}}_{\theta,j,c}^1 \,, \tag{1}$$

where $\tilde{\boldsymbol{p}}_\theta^1 = \text{softmax}(\tilde{\boldsymbol{z}}_\theta^1 \cdot \mathcal{C}_\theta / \tau_s)$ and $\boldsymbol{q}_\xi^2 = \text{softmax}(\boldsymbol{z}_\xi^2 \cdot \mathcal{C}_\xi / \tau_t)$ are the cluster assignments of the student and teacher models respectively, $\tau_s$ and $\tau_t$ are temperature parameters of the student and teacher models respectively, and $\mathcal{P}$ is the set of matched patch pairs between views.

**Object-level constrastive learning.** Now that we have a set of object features that are aligned between views, we can apply a contrastive learning objective to perform object-centric learning. Not all slots are used. We only keep the slots that at least occupy one patch and we filter out the redundant ones by computing the following binary indicator: $\mathbb{1}_i = \exists_j$ such that $\text{argmax}_c(p_\theta(\boldsymbol{v}_j^1)_c) = i$. Those with the same tokens form positive pairs and others form negative pairs. We adopt a MoCo-style contrastive learning approach. Let $\boldsymbol{s}_{\theta,i}^1 = \sum_j p_\theta(\boldsymbol{v}_j^1)_i \boldsymbol{z}_{\theta,j}^1$ and $\boldsymbol{s}_{\xi,i}^2 = \sum_j q_\xi(\boldsymbol{v}_j^2)_i \boldsymbol{z}_{\xi,j}^2$ be the slots from the student and teacher models respectively. The contrastive loss is defined as:

$$\mathcal{L}_{\text{slot}}(\tilde{\boldsymbol{s}}_\theta^1, \boldsymbol{s}_\xi^2) = -\frac{1}{K} \sum_{i=1}^{C} \log \frac{\mathbb{1}_i^1 \mathbb{1}_i^2 \exp(h_\theta(\boldsymbol{s}_{\theta,i}^1) \cdot \boldsymbol{s}_{\xi,i}^2 / \tau)}{\sum_{j=1}^{C} \mathbb{1}_i^1 \mathbb{1}_j^2 \exp(h_\theta(\boldsymbol{s}_{\theta,i}^1) \cdot \boldsymbol{s}_{\xi,j}^2 / \tau)} \,, \tag{2}$$

where $h_\theta$ is a predictor MLP, $K = \sum_i \mathbb{1}_i^1 \mathbb{1}_i^2$ is the number of positive pairs and $\tau$ is a temperature parameter set to $0.2$ following Chen et al. (2021). $\ell_2$-normalization is applied to both slots and their predictions before computing the inner product. The final loss is a combination of these objectives:

$$\mathcal{L}_{\theta,\xi}(\tilde{\boldsymbol{v}}^1, \boldsymbol{v}^2) = \lambda_1 \mathcal{L}_{\text{patch}}^{\text{within}}(\tilde{\boldsymbol{v}}^1, \boldsymbol{v}^2) + \lambda_1 \mathcal{L}_{\text{patch}}^{\text{cross}}(\tilde{\boldsymbol{v}}^1, \boldsymbol{v}^2) + \lambda_2 \mathcal{L}_{\text{slot}}(\tilde{\boldsymbol{s}}_\theta^1, \boldsymbol{s}_\xi^2) \,, \tag{3}$$

where $\mathcal{L}_{\text{patch}}^{\text{within}}$ is exactly the same as $\mathcal{L}_{\text{iBOT}}$ and $\lambda_1 = 0.5$ and $\lambda_2 = 1$ are weighting coefficients. In practice the symmetrized objective $\mathcal{L}_{\theta,\xi}(\tilde{\boldsymbol{v}}^1, \boldsymbol{v}^2) + \mathcal{L}_{\theta,\xi}(\tilde{\boldsymbol{v}}^2, \boldsymbol{v}^1)$ is optimized.

**Connection to previous work.** As shown in Tab. 1, SlotMIM shares key ideas with previous self-supervised learning methods. MoCo v3 (Chen et al., 2021) and DINO (Caron et al., 2021) perform instance discrimination on image crops. SlotCon (Wen et al., 2022) performs contrastive learning between slots, but its objectness only receives high-level signals. This worked well for ResNet since the network architecture provided strong inductive bias for objectness. But when applied to ViT, MIM-like low-level signal is needed. Regard-

Table 1: Holistic comparison with previous methods.

| Method | mask | $\mathcal{L}_{\text{patch}}^{\text{cross}}$ | $\mathcal{L}_{\text{patch}}^{\text{within}}$ | $\mathcal{L}_{\text{slot}}$ | $k$-NN | ADE | Jacc | Loc |
|---|---|---|---|---|---|---|---|---|
| MoCo v3 | ✗ | ✗ | ✗ | △ | 43.3 | 41.3 | – | – |
| DINO | ✗ | ✗ | ✗ | ○ | **46.3** | 40.5 | – | – |
| SlotCon | ✗ | ✓ | ✗ | ✓ | 42.9 | 47.1 | 40.1 | 59.6 |
| iBOT | ✓ | ✗ | ✓ | ○ | 45.3 | 44.5 | – | – |
| SlotMIM | ✓ | ✓ | ✓ | ✓ | 46.2 | **49.1** | **43.9** | **62.5** |

△: contrastive learning on image crops
○: self-distillation on image crops

ing iBOT, its patch-level and global self-distillation loss are built on the same set of prototypes, which however requires different levels of complexity. Our design allows the prototypes to focus on learning fine-grained patterns for patch-level loss, and semantic learning is achieved by other modules.

# 3 EXPERIMENTS

## 3.1 SETTING

Table 2: **Overview of pre-training datasets.** We uniformly sample subsets of 241K[1]images from ImageNet, CC12M, and Ego4D. COCO+ is formed by combining `train` and `unlabeled` subsets of COCO. For Ego4D we first extract frames at 0.2 fps and then sample image subsets.

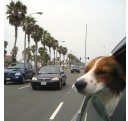
Object-centric

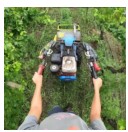
Scene-centric

| Pre-train Data | Source | #Image | #Obj/Img | #Class | Type | Video |
|---|---|---|---|---|---|---|
| INet-241K | ImageNet | 241K | 1.7 | 1000 | OC | ✗ |
| COCO+ | COCO | 241K | 7.3 | 80 | SC | ✗ |
| CC-241K | CC12M | 241K | – | – | Web | ✗ |
| Ego-241K | Ego4D | 241K | – | – | Ego | ✓ |

OC: Object-centric; SC: Scene-centric; Web: Web-crawled; Ego: Ego-centric

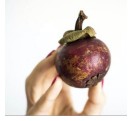
Web-crawled

Ego-centric

**Dataset.** We consider pre-training on four types of datasets, including object-centric ImageNet (Deng et al., 2009), scene-centric COCO (Lin et al., 2014), web-crawled CC12M (Changpinyo et al., 2021), and ego-centric Ego4D (Grauman et al., 2022). For the baseline setting, we uniformly sample 241K images from each dataset to form the training sets. See Tab. 2 for details. For larger-scale pre-training, we sample 1.28M images from the same sources, except for scene-centric data where we switch to Open Images (Kuznetsova et al., 2020).

**Methods.** We compare with a variety of ViT pre-training methods, including BEiT (Bao et al., 2022), SplitMask (El-Nouby et al., 2021), MAE (He et al., 2022), DINO (Caron et al., 2021), iBOT (Zhou et al., 2022), and DINOv2 (Oquab et al., 2024). We train with official code and suggested hyperparameters. For DINO and iBOT, training instability is observed when training on NOC data, and we tuned the teacher temperature if necessary for convergence.

**Pre-training setting.** We use ViT-B/16 (Dosovitskiy et al., 2021) as the backbone. At 241K data scale, all methods are trained for 800 epochs by default. At 1.28M data scale, we train for 400 epochs. The optimization hyperparameters follow Zhou et al. (2022).

**Evaluation setting.** We evaluate models on ImageNet-1K (Deng et al., 2009) and ADE20K (Zhou et al., 2017) following He et al. (2022). For ImageNet linear probing, we sweep between `[CLS]` token and average pooling and report best results of each model. For ImageNet fine-tuning, all models use the average-pooled token. Under both settings, we report top-1 validation accuracy of a single $224 \times 224$ center crop. ADE20K semantic segmentation experiments use UperNet (Xiao et al., 2018) and train for 160K iterations with batch size 16. Additionally, COCO object detection and instance segmentation is also considered to evaluate the transferability of pre-trained models. We follow the

---

[1]1.28M subsets are also considered. For scene-centric data, we use Open Images dataset to scale up.

same setting in (Zhou et al., 2022) to train a Cascade Mask R-CNN (Cai & Vasconcelos, 2019) with $1\times$ schedule (12 epochs), and report box and mask AP.

**Analytical metrics.** We also introduce some numeric indicators to help analyze some properties of pre-trained models. This includes $k$-NN ImageNet classification ($k = 20$) following Caron et al. (2021), and object discovery metrics evaluated on Pascal VOC 2012 following Venkataramanan et al. (2024); Siméoni et al. (2021). Jaccard similarity measures the overlap between predicted mask $P$ and the ground truth mask $G$ as $J(P, G) = \frac{G \cap P}{G \cup P}$. We also compute CorLoc, which measures the percentage of correctly located boxes, where a predicted box is correct if it's IoU $\geq 0.5$. Additionally, we maintain a running mean of the average number of active slots $\overline{K}$ in an image during training.

## 3.2 RESULTS UNDER BASELINE PRE-TRAINING BUDGET

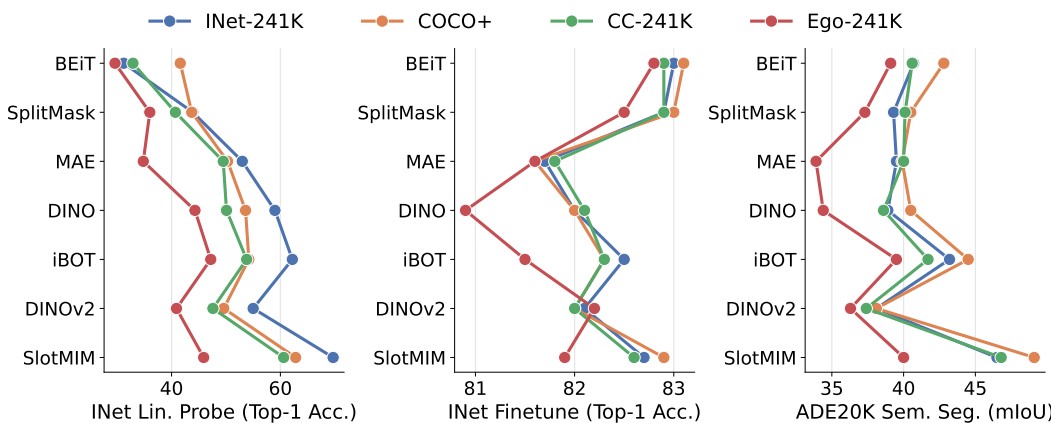

Figure 3: **Different models learn different levels of information from different datasets.** SlotMIM consistently outperforms prior arts whether pre-trained on object-centric data or not. Notably, when trained on COCO+, it transfers better than most ImageNet models despite the domain gap (middle). When evaluated on segmentation, the superiority of our method is even more pronounced (right).

We first evaluate models pre-trained on 241K-scale datasets, and show that NOC data can be good learning resources if used properly. The results are present in Fig. 3. Overall, SlotMIM achieves the best performance across classification and segmentation tasks, no matter learning from object-centric data or not. Below, we discuss some other interesting findings.

**Features learned from NOC data can be linear separatable on ImageNet.** From Fig. 3 (left), our models trained on COCO and CC achieve similarly good linear probing performance on ImageNet with best prior ImageNet-trained methods. As a clear contrast, all previous methods trained on NOC datasets (COCO, CC, and Ego4D) fall behind best ImageNet counterpart.

**NOC data can be worth more than ImageNet for ImageNet.** As shown in Fig. 3 (middle), under ImageNet fine-tuning setting, the top-3 methods (BEiT, SplitMask, and SlotMIM) have best performance when trained on COCO+ instead of ImageNet. For MAE and DINO, training on CC also transfers better than ImageNet. Note that this is uncommon given the domain gap between NOC pre-training data and OC downstream task, demonstrating that NOC data are information-rich learning resources.

**NOC data is significantly beneficial for similar-domain downstream tasks.** In Fig. 3 (right), we evaluate the models on ADE20K semantic segmentation. SlotMIM trained on COCO+ achieves the best performance, and our CC and ImageNet-trained models also surpass prior models by a large margin. This suggests that NOC data can be particularly useful for scene-understanding tasks.

**Ego-centric data solely is not suitable for general-purpose models.** In Fig. 3, we observe that models trained on Ego4D generally perform worse than those trained on other datasets. This is possibly due to video-based ego-centric data's redundancy and suggests that data diversity matters more for general-purpose pre-training. Still, as will be discussed in Sec. 3.5, ego-centric data can be effective for robot learning and SlotMIM learns the best representations from it.

## 3.3 SCALING UP PRE-TRAINING SCHEDULE

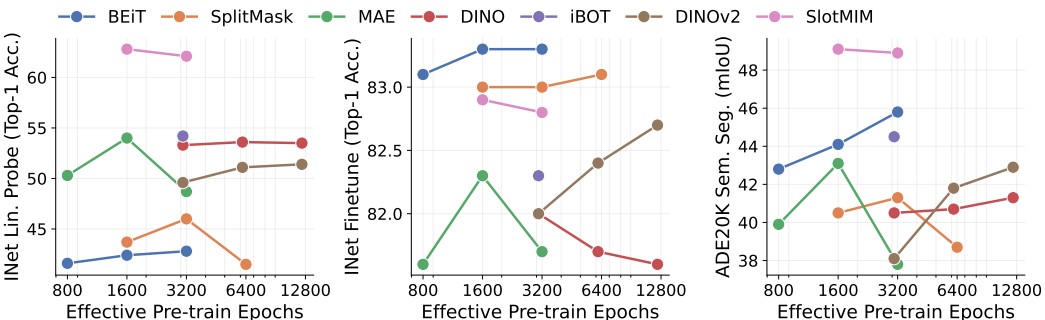

Figure 4: **A convergence check on COCO+ with longer training.** Existing methods either experience performance degradation or stagnate, or require significantly more epochs to reach better performance. SlotMIM achieves leading performance with shorter training (also without multi-crop).

SlotMIM is not only efficient in the need of data scale, but also in the need of training epochs. We take training on COCO+ as an example, and compare SlotMIM with other methods considering longer training schedules. For a fair comparison, we follow previous literature (Zhou et al., 2022) to calculate effective pre-training epochs for each method, which is $3.84\times$ for methods using multi-crop (DINO, iBOT, and DINOv2), $2\times$ for contrastive methods including SlotMIM, and $1\times$ for non-contrastive methods (*e.g.*, BEiT and MAE). The results are shown in Fig. 4. We observe that SlotMIM achieves the best performance with the shortest training schedule, and other methods either require significantly more epochs to reach better performance or experience performance degradation or stagnation.

## 3.4 SCALING UP PRE-TRAINING DATA

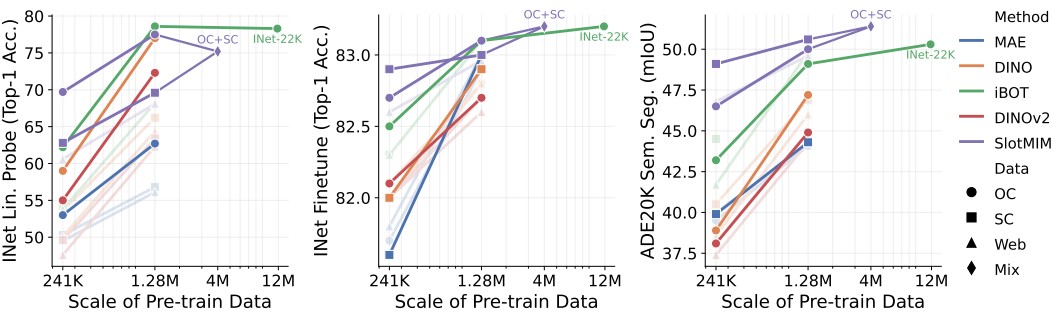

Figure 5: **Scaling laws on different data sources.** We scale up object-centric, scene-centric, and web-crawled data, and highlight the best (model, data) combinations. Our method learns strong and transferable representations with significant data efficiency and continues to improve with more data.

Superior data efficiency allows us to explore larger-scale pre-training data. In Fig. 5, we show that SlotMIM achieves strong performance with remarkable data efficiency.

**Comparable or better performance with small data scale.** As shown in Figure 5, SlotMIM achieves comparable or superior performance to other methods using significantly less data. Our INet-241K model for ImageNet linear probing, and COCO+/INet-241K models for ImageNet finetuning and ADE20K semantic segmentation outperform or match most models trained on 1.28M ImageNet images across various tasks. This remarkable data efficiency demonstrates our approach's effectiveness in extracting rich, transferable features from limited data.

**NOC pre-training rivals ImageNet pre-training for ImageNet.** Interestingly, we observe that pre-training on NOC datasets like OpenImages-1.28M can lead to performance better than pre-training on ImageNet for the ImageNet classification task (fine-tuning setting). When scaled up to 4M scale, this trend becomes more pronounced. This aligns with the trend in Fig. 3 that NOC data can provide more information-rich features, which can be better-utilized by models like SlotMIM.

**NOC data also possesses stronger scalability.** We extend experiments to the 4M scale by combining INet-1.28M (Deng et al., 2009), COCO+ (Lin et al., 2014), OpenImages (Kuznetsova et al., 2020), Objects365 (Shao et al., 2019), and LVIS (Gupta et al., 2019b). Compared with previous efforts on scaling up with ImageNet-22K (12M images) (Russakovsky et al., 2015), the performance of our SlotMIM models continues to improve and surpasses them with $3\times$ less data. This suggests that NOC data can be a more scalable learning resource.

### 3.4.1 OBJECT DETECTION AND INSTANCE SEGMENTATION

Figure 6: **Transfer learning experiments on COCO object detection and instance segmentation.** SlotMIM shows better data efficiency with both OC and NOC data, and the performance continually grows with more data, surpassing all prior SoTA models by a notable margin.

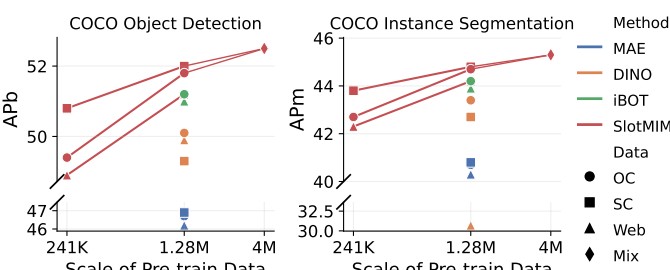

In Fig. 6, we also present an evaluation on COCO object detection and instance segmentation. The superiority of SlotMIM is clear and remains improving with increased data scale.

### 3.5 PRE-TRAINED VISION MODELS FOR MOTOR CONTROL

Previous sections showed that 1) NOC data offers rich, transferable features for image recognition and scene understanding tasks, and 2) its advantages are especially evident when there is strong alignment between pre-training data and target downstream tasks. In this section, we analyze the effects of OC/NOC data types (ego-centric and scene-centric) on robot manipulation benchmarks and the data efficiency of SlotMIM.

Table 3: **Overview of robot manipulation tasks.** Right: example tasks of each benchmark suite.

| Bench. Suite | RGB | Proprio. | #Task | #Demo | #Seed |
|---|---|---|---|---|---|
| Franka Kitchen | ✓ | ✗ | 5 | 25 | 3 |
| Meta-World | ✓ | ✗ | 8 | 25 | 3 |

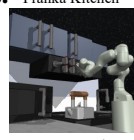 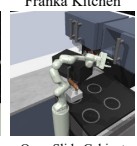 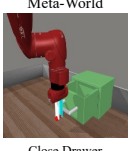 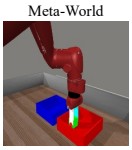

Franka Kitchen — Turn on Knob    Franka Kitchen — Open Slide Cabinet    Meta-World — Close Drawer    Meta-World — Pick Bin

**Imitation learning setups.** Following Hu et al. (2023), we compared our methods across two robot manipulation benchmarks using behavior cloning: Franka-kitchen (Gupta et al., 2019a) and Meta-world (Yu et al., 2019). We focus on efficient real-world learning with behavior cloning (BC) using a few human demonstrations per task in each benchmark suite. For each pre-trained vision model and task, we run 3 seeds of BC due to the result's high variability. Detailed setups for behavior cloning and example tasks are shown in Tab. 3. One-image observation for its comparable performance to stacks of images and higher computational efficiency. All tasks and environments use $224\times224$ RGB images without proprioceptive input. No image augmentations, such as random shifts, are applied. The policy training includes a few modifications: The policy network is trained for 20,000 steps, following R3M (Nair et al., 2023). We employ attentive pooling, as in V-Cond (Karamcheti et al., 2023), which is shown to be the better choice than the default `[CLS]` embedding head and provides better comparisons between pre-trained frozen visual representations.

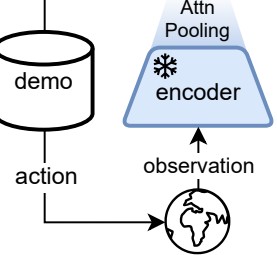

Figure 7: Behavior cloning with attentive probing.

**Baselines.** As shown in Fig. 8, MAE regime (blue line) including MVP (Radosavovic et al., 2023) and VC-1 (Majumdar et al., 2023) that leverages MAE (He et al., 2022) to pre-train the model across a massive collection of ego-centric videos (Grauman et al., 2022) and Internet data. V-Cond (Karamcheti et al., 2023) (purple point) further proposes language-driven representation

learning from human videos and associated captions. DINO (Caron et al., 2021) (orange line) is based on self-distillation and iBOT (Zhou et al., 2022) (green line) further combines MIM with self-distillation.

Figure 8: **Pre-training for robot manipulation tasks.** This evaluation considers three factors that influence manipulation success rates: data types (ego-centric ◆, object-centric ●, and scene-centric ■), pre-training methods, and data scale. Dark lines represent the best-performing data scaling for each pre-training method, while light lines indicate sub-optimal performance.

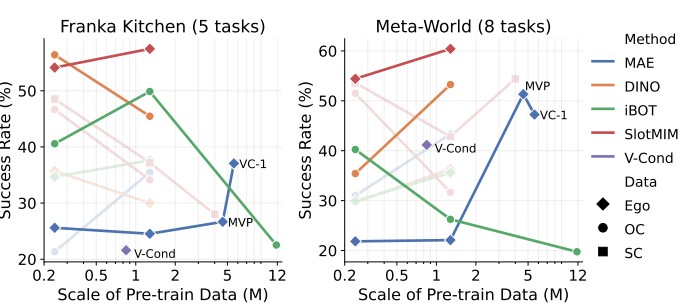

Fig. 8 examines the relationship between manipulation success rates and pre-training methods, comparing the trend of scaling dataset size across different data types: ego-centric ◆, object-centric ●, and scene-centric ■. Notably, increasing dataset size does not always improve performance across benchmarks, as also reported by VC-1(Majumdar et al., 2023).

**Different scaling behaviors of OC/SC vs. ego-centric data**. In object manipulation tasks as shown in Fig. 8, scaling scene-centric and object-centric data to the million level can lead to performance drops for methods like MAE, DINO, and iBOT. We hypothesize that self-supervised representation learning, including MIM, aims to learn invariance, where the feature extractor pulls images with similar visual content together in the embedding space, compressing the visual data. However, scaling up data may result in over-compression, causing performance drops in visuomotor control tasks.

By contrast, using ego-centric data for pre-training, MAE (blue line) and SlotMIM (red line) show positive data scaling effects. Unlike SC/OC data from vast Internet sources, ego-centric images are sampled from consecutive human videos that share contextual backgrounds or scenarios. The ego-centric data are among daily scenarios such as household, outdoor, workplace, and leisure *etc*. that are contextually similar to the robot manipulation scenarios (Grauman et al., 2022). Thus, invariance learning in ego-centric data tends to focus more on the differences within the same video or scenario, particularly in the foreground objects. This focus is critical for robot manipulation learning, as it requires effective interaction with these foreground objects.

**SlotMIM is more data efficient in leveraging ego-centric data.** Compared to general-purpose pre-trained models and state-of-the-art (SoTA) robot learning methods (e.g., MVP (Radosavovic et al., 2023) and VC-1 (Majumdar et al., 2023)), we demonstrate that SlotMIM (dark red ◆ line), pre-trained with just 241K data samples, can surpass prior methods that utilized over 1 million samples. When scaled to 1 million ego-centric data, it achieves the highest success rates compared to all other methods in the figure.

### 3.6 ABLATION STUDY

This section presents an ablation study on key SlotMIM design choices. Models are trained on COCO+ for 800 epochs. Tab. 4 demonstrates the impact of different modules, comparing $k$-NN ImageNet classification, ADE20K semantic segmentation, Jaccard similarity, and CorLoc for object discovery mask quality and localization recall. We

Table 4: **Ablation study on effective modules.**

| | mask | $\mathcal{L}_{\text{patch}}^{\text{cross}}$ | $\mathcal{L}_{\text{patch}}^{\text{within}}$ | $\mathcal{L}_{\text{slot}}$ | $k$-NN | ADE | Jacc | Loc | $\overline{K}$ |
|---|---|---|---|---|---|---|---|---|---|
| 1 | ✗ | ✓ | ✗ | ✗ | 45.1 | 47.4 | 42.5 | 55.6 | 8.3 |
| 2 | ✓ | ✓ | ✗ | ✗ | 44.9 | 48.6 | 42.3 | 60.7 | 10.3 |
| 3 | ✓ | ✗ | ✓ | ✗ | 27.7 | 45.7 | 39.3 | **65.5** | 20.7 |
| 4 | ✓ | ✓ | ✗ | ✓ | 45.3 | 47.5 | 42.9 | 63.6 | 8.4 |
| 5 | ✓ | ✓ | ✓ | ✓ | **46.2** | **49.1** | **43.9** | 62.5 | 9.4 |

report the average number of objects/stuff discovered per image. Results show Jaccard similarity correlates with representation quality, suggesting better object discovery improves representation learning. Introducing MIM enhances object localization and benefits segmentation tasks (rows 1 and 2). Cross-view consistency and slot contrastive losses contribute to improved object discovery (rows 2, 3, 5). The within-view loss can serve as a regularizer and improve representations (rows 4, 5).

Table 5: **Ablation studies on hyperparameters.** Default values are marked with a  cyan  background.

| (a) **Number of prototypes** | | | | | | (b) **Mask ratio** ($\pm$**0.2**) | | | | | | (c) **Patch loss** | | | | | |
|---|---|---|---|---|---|---|---|---|---|---|---|---|---|---|---|---|---|
| $C$ | $k$-NN | ADE | Jacc | Loc | $\overline{K}$ | | $k$-NN | ADE | Jacc | Loc | $\overline{K}$ | Type | $k$-NN | ADE | Jacc | Loc | $\overline{K}$ |
| 256 | 45.3 | **49.1** | 42.2 | 61.2 | 7.8 | 0.3 | **46.2** | **49.1** | 43.9 | 62.5 | 9.4 | center | **46.2** | 49.1 | **43.9** | 62.5 | 9.4 |
| 512 | **46.2** | **49.1** | **43.9** | **62.5** | 9.4 | 0.4 | 45.8 | 48.6 | 45.0 | 62.6 | 8.1 | SH | 45.1 | **49.3** | 40.8 | **68.5** | 15.2 |
| 1024 | 45.6 | 48.4 | 42.8 | 62.6 | 10.8 | 0.5 | 44.3 | 48.2 | **45.7** | 64.8 | 7.1 | | | | | | |

In Tab. 5 we present ablations on some numeric design choices. Generally speaking, a smaller number of prototypes, a higher mask ratio, and the use of centering (Caron et al., 2021) instead of Sinkhorn-Knopp algorithm (Caron et al., 2020) encourage the network to discover more holistic concepts/objects, while the opposite discovers more fine-grained ones. Optimal representation is highly related to object discovery quality.

## 4 RELATED WORK

**Self-supervised representation learning.** Self-supervised representation learning aims to extract transferrable features from unlabeled data (Tian et al., 2020a; Caron et al., 2018; 2020; 2021; Asano et al., 2020; Chen et al., 2020). Two primary approaches have emerged: contrastive learning (Tian et al., 2020a; Chen et al., 2020; He et al., 2020), which learns by comparing positive and negative examples, and masked image modeling (He et al., 2022; Xie et al., 2022), which reconstructs masked regions of images. While these methods have shown success, they've primarily been tested on object-centric datasets like ImageNet-1K. Our study extends this by exploring self-supervised learning on large-scale non-object-centric datasets, demonstrating superior data efficiency compared to previous pre-training methods across various downstream applications. Additionally, we provide insights into the scalability and generalizability of these methods across diverse data types.

**Learning on non-object centric data.** Recent works have addressed the challenge of self-supervised learning on non-object centric data (Van Gansbeke et al., 2021; Oquab et al., 2024; Xie et al., 2021; Wang et al., 2021; Hénaff et al., 2021; 2022). These efforts include dense contrastive learning approaches (Wang et al., 2021; Xie et al., 2021), object-centric methods for dense prediction tasks (Hénaff et al., 2022), and slot-based contrastive learning frameworks (Wen et al., 2022). Additionally, some methods focus on learning from uncurated datasets (Caron et al., 2019; Tian et al., 2021; Bai et al., 2022). In our work, we decompose object representations to leverage established techniques that enable fine-grained pattern learning through patch-level target design, facilitating effective pre-training.

**Scaling vision pre-training.** Scaling vision pre-training to larger datasets and models has become a significant focus in recent years (Tian et al., 2021; Caron et al., 2019; Mu et al., 2022; Radford et al., 2021; Dehghani et al., 2023; Gadre et al., 2023; Schuhmann et al., 2022). The creation and use of massive datasets like LVD-142M (Oquab et al., 2024) and LAION-5B (Schuhmann et al., 2022) have also played a crucial role. Our method examines how non-object-centric datasets influence data scaling and the transferability of learned representations across downstream tasks, focusing on fine-grained data types: object-centric, scene-centric, ego-centric, and mixed.

## 5 CONCLUSION

This work revisits the use of non-object-centric (NOC) data for self-supervised visual representation learning. Our comprehensive study demonstrated that NOC data holds immense potential due to its rich information, which has been largely underutilized. To harness this potential, we formalized learning from NOC data into two sub-tasks: scene decomposition and object-centric representation learning. By repurposing and integrating established techniques to target these sub-tasks, we developed SlotMIM, a unified framework capable of effectively handling both NOC and object-centric data. Through extensive experiments across diverse datasets and downstream tasks, including robotics, we demonstrated the consistent superiority of our approach over existing methods. We hope our promising results open new avenues for scaling self-supervised learning using large volumes of NOC data, overcoming the limitations posed by conventional datasets in representation learning.

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

# A APPENDIX

## A.1 IMPLEMENTATION DETAILS

### A.1.1 PRE-TRAINING

**Architecture.** We use ViT-B/16 (Dosovitskiy et al., 2021) as our backbone. The projector $g$ and predictor $h$ are 3-layer MLPs with hidden dimension 4096 and output dimension 256.

**Optimization.** We use AdamW optimizer with a cosine learning rate schedule, peak learning rate of 1.5e-4, and weight decay of 0.05. The learning rate is linearly ramped up during the first 10 epochs to its base value scaled with the total batch size: $lr = lr_{\text{base}} \times \text{batch size}/256$. We train for 800 epochs on 241K-scale datasets and 400 epochs on 1.28M-scale datasets, with a batch size of 1024 distributed across 8 A100 GPUs. For experiments on 4M-scale datasets, we train 200 epochs.

**Augmentation and masking.** We use the same augmentation strategy as in iBOT (Zhou et al., 2022) except not using small local crops. The masking strategy follows (Zhou et al., 2022), with prediction ratio $r$ uniformly sampled from range $[0.3 - 0.2, 0.3 + 0.2]$.

**Hyperparameters.** We set $\tau_s = 0.1$, $\tau_t = 0.07$. The number of prototypes is set to 512 for COCO and 1024 for other datasets.

### A.1.2 EVALUATION

**Linear probing and fine-tuning on ImageNet-1K.** We follow (He et al., 2022) for details on ImageNet evaluations. For linear probing, we insert an extra BatchNorm layer without affine transformation between the features and the linear classifier. We train with batch size 4096, initial learning rate 0.1, and optimize using SGD for 90 epochs. We sweep between `[CLS]` token and average pooling and report the best results of pre-trained models. For fine-tuning, We train a linear classifier on frozen features for 100 epochs using SGD with momentum 0.9, batch size 1024, and initial learning rate 1e-3 with cosine decay. For both settings, accuracy is evaluated on a single 224×224 crop.

**Semantic segmentation on ADE20K.** We use UperNet Xiao et al. (2018) implemented in MMSegmentation following Zhou et al. (2022). Specifically, we fine-tune for 160k iterations with stochastic gradient descent, with a batch size of 16 and weight decay of 0.0005. The learning rate is 0.01 and decays following the poly schedule with power of 0.9 and min_lr of 0.0001.

**Object detection and instance segmentation on COCO.** COCO object detection and instance segmentation setting also follows Zhou et al. (2022), where the pre-trained model initialized a Cascade Mask R-CNN (Cai & Vasconcelos, 2019). The image scale is [640, 800] during training and 800 at inference. We fine-tune all layers end-to-end on COCO Lin et al. (2014) `train2017` set with the standard 1× schedule and report AP for boxes and masks on the `val2017` set.

**Robot manipulation tasks.** Following the setup of Hu et al. (2023), the policy network of behavior cloning includes a LayerNorm layer before the MLP. The policy training involves mini-batches of 128 samples, conducted over 20,000 steps with the Adam optimizer set to a learning rate of 0.0001. For each pre-trained vision model and task, we run 3 seeds of BC due to the result's high variability. One-image observation for its comparable performance to stacks of images and higher computational efficiency. All tasks and environments use 224×224 RGB images without proprioceptive input. No image augmentations, such as random shifts, are applied. We employ attentive pooling, as in V-Cond (Karamcheti et al., 2023), which is shown to be the better choice than the default `[CLS]` embedding head and provides better comparisons between pre-trained frozen visual representations.