# OpenReview forum: "An Image is Worth $K$ Slots: Data-efficient Scaling of Self-supervised Visual Pre-training"
_ICLR.cc/2025/Conference — ICLR 2025 Conference Withdrawn Submission_

### Official Review · Reviewer_qX11 · 2024-10-19

**Soundness:** 2
**Presentation:** 3
**Contribution:** 2
**Rating:** 5
**Confidence:** 3

**Summary:**

The paper introduces a method to improve scaling up visual encoder training using non-object-centric data. This paper proposes a semantic bottleneck in Masked Image Modeling (MIM) to enhance objectness at the patch-level token representation. Additionally, this paper implements cross-view consistency regularization to promote multiview invariance, facilitating semantic object discovery and enabling instance discrimination between object-level features. The authors validate their approach across various datasets, including object-centric, scene-centric, web-crawled, and ego-centric data, demonstrating significant improvements in image recognition, scene understanding, and robot learning.

**Strengths:**

1. The paper is written clearly and effectively expresses its findings and methodology.
2. The authors conduct a thorough evaluation across multiple datasets and tasks, showcasing the robustness of their approach.
3. The application of the method in robot learning contexts adds practical significance to the research.

**Weaknesses:**

1. The novelty of this work requires further justification. The approach is built iBOT and combines several existing techniques, such as iBOT and such as slot attention and cross-view contrast, without presenting substantial new contributions.
2. The categorization of datasets into object-centric, scene-centric, and non-object-centric may be oversimplified. For instance, the scene-centric dataset COCO can contain simple scenes with clear foregrounds.
3. It's difficult to determine whether the difference in model performance on different datasets comes from being object-centric or not, or other data distribution influences, including but not limited to object class distribution, image quality distribution, image resolution, etc.
4. The validations are still conducted on the previous datasets, making it difficult to determine how the training data affects the test data with different number distributions.

**Questions:**

Please refer to weaknesses.

---

### Official Review · Reviewer_oBHS · 2024-11-03

**Soundness:** 2
**Presentation:** 1
**Contribution:** 1
**Rating:** 3
**Confidence:** 4

**Summary:**

The paper describes a technique for pre-training visual encoders that is specifically designed for non-object centric data, e.g. complex scenes or egocentric videos.
In doing so, the authors also train and evaluate previous models to characterize the impact of data in the pre-training process.
For its architecture, the paper borrows key modules and losses from existing literature, combining them in a way to improve semantic object understanding.
The experiments focus on both demonstrating the effectiveness of the proposed method and on discussing the effect of data.

**Strengths:**

The paper aims to shine a light on the role of data in the context of self-supervised pre-training, with an emphasis on downstream performance.
In the language modeling literature, the importance of data is well-established, but in the vision community, the focus has been on model architectures and training objectives. Therefore, the paper addresses an important gap in the literature.

At the same time, the paper puts forward a new pre-training method SlotMIM which combines several existing components.

However, the paper seems to mix together two objectives: studying the effect of data and proposing a new pre-training method.
As a consequence, the paper lacks a clear narrative and the contributions are not well separated.
A practical example of this issue are the chaotic plot lines in figures 5, 6 and 8.

On the evaluation side, the authors experiment with a variety of tasks, from classification to segmentation and robotics control.
The diversity of the tasks is a strength of the paper, as it allows to assess the generalization of the proposed method and the importance of data in different contexts.

**Weaknesses:**

**Object-centric terminology:**
Throughout the paper, the term "object-centric" is used to characterize datasets like ImageNet, whose images generally include a single object, and to contrast them to OpenImages, SA-1B, etc. that contain multiple objects.
This terminology is not standard and may be confusing to readers, for example on L57 "our study reveals that several insights from object-centric learning remain applicable to NOC data", where "insights from object-centric learning" are derived from DINO, IBOT, SimCLR, etc. trained on ImageNet.
Also, many related works describe datasets like COCO as "object-centric" and claim that ImageNet is not "object-centric" enough to train object-centric models on, which is the opposite of the terminology used in this paper.
To avoid the confusion, the authors could consider using terms like "single-object" or "multi-object" datasets instead.

**Other terminology:**
Other aspects of the chosen terminology are also non-standard and confusing. A few examples:
- L120 "normalizes attention scores on the query side instead of the key side" (axis?)
- L152 "assigns each patch token a soft one-hot encoding" (can't be both soft and one-hot)

**Vague claims:**
The paper makes several claims that are not supported by the experiments or existing literature. For example:
- L182-185 "A key factor contributing to the lack of semantic meaning is that the iBOT loss LiBOT is computed between patches within the same view. Consequently, there is no explicit guidance for learning invariant representations across different views of the same object or scene."
  However, self-supervised works like I-JEPA (https://arxiv.org/pdf/2301.08243) demonstrate that predicting missing patches in latent space leads to learning semantically meaningful representations.
- L224-226 "This worked well for ResNet since the network architecture provided strong inductive bias for objectness."
  What are the inductive biases towards objects in a convolutional network? On the contrary, previous papers like Slot Attention (https://arxiv.org/abs/2006.15055) build on the idea that CNNs lack such biases and proposed to add them explicitly.
- L533-535 "To harness this potential, we formalized learning from NOC data into two sub-tasks: scene decomposition and object-centric representation learning."
  The experiments in the paper do not demonstrate the effectiveness of this formalization in sub-tasks, nor that the proposed architecture and loss implement the sub tasks separately.

**Limited novelty and missing comparison with previous work:**
The loss formulation with cross-view patch alignment is similar to that of VicRegL (https://proceedings.neurips.cc/paper_files/paper/2022/file/39cee562b91611c16ac0b100f0bc1ea1-Paper-Conference.pdf), which is not cited or compared against in the paper.
Furthermore, most components are simply imported from existing work:
- The patch-wise loss is that of IBOT
- The strategy of pooling patch features based on prototype assignments is from SlotCon
- The contrastive loss for the slots is again from SlotCon and MoCo
Undoubtedly, a novel combination of such components and a thorough analysis of their effect would be valuable, but the paper lacks in this aspect.

**Insufficient evaluation:**
L310-312 "under ImageNet fine-tuning setting, the top-3 methods (BEiT, SplitMask, and SlotMIM) have best performance when trained on COCO+ instead of ImageNet."
The gap in performance is around 0.1% which is within margin of random fluctuations, no conclusions can be drawn from this.
If the authors want to provide a convincing argument, all experiments should be repeated multiple times and reported with confidence intervals.

**Evaluation protocol:**
A few choices in the evaluation protocol are questionable:
- L264-265: "For ImageNet fine-tuning, all models use the average-pooled token."
  This is not standard practice: if a model is trained with a CLS token the expectation is that the best global representation for a classification task is in the CLS token, not the average-pooled token.
- All pre-training datasets are limited to 241k images, which is a small fraction of what those models are usually trained on, and for which their training recipes are optimized. From the ImageNet linear probe results in figure 3, all the models seem undertrained, which makes all comparisons invalid.

**Questions:**

**Evaluation results:**
Would it be possible to evaluate the official checkpoints of all third-party methods mentioned in the paper?
This would remove any potential issue with implementation and under training.

**Proposed method:**
Can you clarify the novel technical components of SlotMIM that set it apart from previous works?

**Figure 1:**
The figure is not very informative because the caption does not explain what each image represent. Please clarify:
- On the left side, how is the visualization obtained, what do the colors represent?
- On the center and right side, what are the queries for the retrieved images? How are they chosen?

**Suggestion:**
Lines L54-57 "we begin by conducting a comprehensive evaluation of existing self-supervised learning approaches on four datasets: object-centric (Deng et al., 2009) and non-object-centric (Lin et al., 2014; Changpinyo et al., 2021; Grauman et al., 2022)"
Could you add the names of the datasets in the text to make it easier to read?

**Incorrect DINO loss:**
On L109, the DINO loss is missing the sharpening and centering operations for the teacher's logits, which are crucial for the method to work.

---

### Official Review · Reviewer_bdyq · 2024-11-04

**Soundness:** 3
**Presentation:** 2
**Contribution:** 2
**Rating:** 3
**Confidence:** 4

**Summary:**

The paper considers the use of non object-centric (NOC) data self-supervised visual pre-training. Studying existing methods on three different NOC datasets, the paper observes that NOC data is not as good as object-centric (OC) data but some insights from existing methods still apply for NOC. With this observation, the paper proposes breaking NOC images into slots corresponding to objects before applying existing self-supervised techniques for object-centric data on these slots. Building on top of iBOT, the paper also introduces several modifications such as cross-view patch consistency regularization and smaller number of prototypes to add more semantic meaning to patches.

The paper shows that the proposed model -- SlotMIM -- performs better than existing self-supervised pre-training techniques on several both OC and NOC datasets. SlotMIM also improves with more data, showing its scalability.

**Strengths:**

The paper provides an interesting study on the impact of pre-training data to existing SSL techniques on three different NOC datasets. This study provides several important, though some of them unsurprising (see weaknesses), insights about the use of NOC data for SSL.

The proposed SlotMIM leads to better performance than competitors on benchmarks, showing its effectiveness.

**Weaknesses:**

The paper's novelty is limited.
  - Many insights in the paper have been discussed in previous work. For example:
    + "Features learned from NOC data can be linearly separable on ImageNet": Many works such as SEER[1], DINOv2 [2] (Table 2, uncurated data) have shown that pre-training on web data leads to features that perform well on ImageNet classification.
    + "NOC data is significantly beneficial for similar-domain downstream tasks": DINOv2 (Table 2) shows very good ADE20K performance with uncurated web dataset, Tolan et al. [3] shows DINOv2 trained on satellite images are better than those trained on ImageNet for an application on satellite images.
    + "non-object-centric data is rich in information with vast potential": a list of works showing good results with NOC data including SEER[1], AIM[4], SlotCoN, Vo et al. [5], ...
  - The proposed SlotMIM is mostly a combination of iBOT (masked patch prediction) and SlotCon (cross view consistency + slot constrast)

Insufficient and incomplete empirical results
  - Good SSL features have to work well on a wide range of downstream tasks and data domain. The paper considers only two benchmarks (ImageNet classification and ADE20K segmentation) to draw conclusions in Secs 3.2-3.4. Some fine-grained, retrieval or ood benchmarks should be included to have a more complete evaluations of the pre-trained features.
  - Missing data points in Figure 5, 6 and 8: performance of features trained with DINO, DINOv2 and MAE on largest-scale datasets should be included to have the full pictures.
  -  The paper lacks a comparison to SlotCon. The performance of SlotMIM without masking and within patch loss should be added to Table 4.

[1] Self-supervised Pretraining of Visual Features in the Wild. Goyal et al., 2021.

[2] DINOv2: Learning Robust Visual Features without Supervision. Oquab et al., 2024

[3] Very high resolution canopy height maps from RGB imagery using self-supervised vision transformer and convolutional decoder trained on aerial lidar. Tolan et al., 2024.

[4] Scalable Pre-training of Large Autoregressive Image Models. El-Nouby et al., 2024

[5] Automatic Data Curation for Self-Supervised Learning: A Clustering-Based Approach. Vo et al., 2024

**Questions:**

Some other comments and questions:
  - L. 201: It is not clear what is p, is it the same as \tilde{p} in Eq. 1?
  - L.201: The definition of \mathbb{1}_i is confusing. The whole definition should be put in a quote.
  - L.264: only cls is often used for linear probing, why do we additionally consider average pooling (of patches?) here?
  - L.266: why don't we use linear probing for ADE20K? It would probably better reveal the quality of the pre-trained features.
  - SlotMIM is based on the assumption that images can be broken down to a set of objects. How would SlotMIM work on other NOC data such as satellite images, pdf documents, ... on which this assumption does not hold?

---

### Official Review · Reviewer_67w3 · 2024-11-10

**Soundness:** 3
**Presentation:** 2
**Contribution:** 2
**Rating:** 3
**Confidence:** 4

**Summary:**

The paper explores the challenge of data-efficient self-supervised visual representation learning on non-object centric images. The core idea of the proposed SlotMIM method is to group patch-level image tokens into object-level feature abstractions, referred to as "slots," thereby decomposing non-object centric (NOC) data into object-centric slots so that object-centric techniques can be effectively applied.

**Strengths:**

- Visual representation learning is a fundamental research topic in computer vision, and learning with non-object centric and ego-centric images is particularly interesting.

- The paper presents extensive experiments using the proposed methods in various scenarios, including ImageNet, COCO, CC12M, and Ego4D.

**Weaknesses:**

- Novelty: The novelty of the approach seems limited, as it primarily combines slot attention and iBOT.

- Performance: The paper lacks tables comparing the method with state-of-the-art (SOTA) visual representation learning methods. From the plots in Fig. 4, the performance appears poor, with approximately 84% ImageNet top-1 accuracy and around 49% ADE20K segmentation mIoU, although the model may not have been sufficiently trained. Additionally, SOTA representation learning methods, such as EVA, EVA-02, and InternImage, are not compared in the paper.

- Motivation: The paper claims to focus on data-efficient representation learning, yet it uses 241,000 images for training, which is still a large number. Visual representation learning is a data- and computation-intensive domain, and it tends to be winner-take-all, with users only downloading and using the most effective models while ignoring insufficiently trained ones.

- Experiments: If the paper aims to emphasize its object clustering performance, it should report unsupervised object segmentation results.
Clarity: The paper fails to clearly illustrate its differences with DINOv2 and iBOT. A figure comparing the overall framework with related works could be added.

- Writing: The phrase "enhance MIM with cross-view consistency regularization" is unclear; what does "cross-view" mean in this context?

**Questions:**

Please address the above weakness.

**Details Of Ethics Concerns:**

No Ethics Concerns

---

### Note · Authors · 2024-11-15

I have read and agree with the venue's withdrawal policy on behalf of myself and my co-authors.